# From Kepler to Newton:
# Inductive Biases Guide Learned World Models in Transformers

**Ziming Liu** [1]  **Surya Ganguli** [1]  **Andreas Tolias** [1]

## Abstract

Can general-purpose AI architectures go beyond prediction to discover the physical laws governing the universe? True intelligence relies on "world models"—causal abstractions that allow an agent to not only predict future states but understand the underlying governing dynamics. While previous "AI Physicist" approaches have successfully recovered such laws, they typically rely on strong, domain-specific priors that effectively "bake in" the physics. Conversely, Vafa et al. (2025) recently showed that generic Transformers fail to acquire these world models, achieving high predictive accuracy without capturing the underlying physical laws. We bridge this gap by systematically introducing three minimal inductive biases. We show that ensuring **spatial smoothness** (by formulating prediction as continuous regression) and **stability** (by training with noisy contexts to mitigate error accumulation) enables generic Transformers to surpass prior failures and learn a coherent **Keplerian** world model, successfully fitting ellipses to planetary trajectories. However, true physical insight requires a third bias: **temporal locality**. By restricting the attention window to the immediate past—imposing the simple assumption that future states depend only on the local state rather than a complex history—we force the model to abandon curve-fitting and discover **Newtonian** force representations. Our results demonstrate that simple architectural choices determine whether an AI becomes a curve-fitter or a physicist, marking a critical step toward automated scientific discovery.

[1]Stanford University. Correspondence to: Ziming Liu <zmliu1@stanford.edu>, Andreas Tolias <tolias@stanford.edu>.

*Proceedings of the 43rd International Conference on Machine Learning*, Seoul, South Korea. PMLR 306, 2026. Copyright 2026 by the author(s).

## 1. Introduction

Given the broad skills and knowledge demonstrated by foundation models (Brown et al., 2020; Chowdhery et al., 2023; Touvron et al., 2023; Radford et al., 2021; Alayrac et al., 2022; Liu et al., 2023; Zitkovich et al., 2023; Kim et al., 2024; Reed et al., 2022), it is natural to expect that they possess robust internal "world models"—causal abstractions that do not merely predict what happens next (e.g., Kepler's geometric fits), but capture the simple physical mechanisms determining why it happens (e.g., Newton's dynamical laws).

This expectation raises a central question: do world models truly emerge within foundation models? Answering this question is a challenge. These models are highly complex, and the notion of a "world model" is often context-dependent or vaguely defined. Thus, it is useful to study world-model emergence in simple, controlled settings where the ground truth is well understood – for instance, Newtonian physics, in which the governing "world model" reduces to a set of simple differential equations. In this vein, Vafa et al. (2025) used planetary motion as a testbed and found that although a transformer can make highly accurate predictions, gravitational forces fail to emerge in its internal representations, even when a GPT-2-scale transformer is trained on datasets as large as 20B tokens. However, the reason behind the failure remains unclear. The central research question of this paper is thus:

> **Research Question:**
>
> Why do transformers fail to learn the Newtonian world model for planetary motion, and how can we fix this problem?

Answering this is a critical litmus test for the vision of developing 'AI Scientists': if general-purpose architectures cannot recover the simple, known laws of classical mechanics, they are unlikely to be trusted to discover the unknown laws of novel phenomena.

We can gain some insights from the success of "AI physicist" models (Wu & Tegmark, 2019; Brunton et al., 2016; Cranmer et al., 2020; Lemos et al., 2023; Liu & Tegmark, 2021; Liu et al., 2022; 2024; Udrescu & Tegmark, 2020),

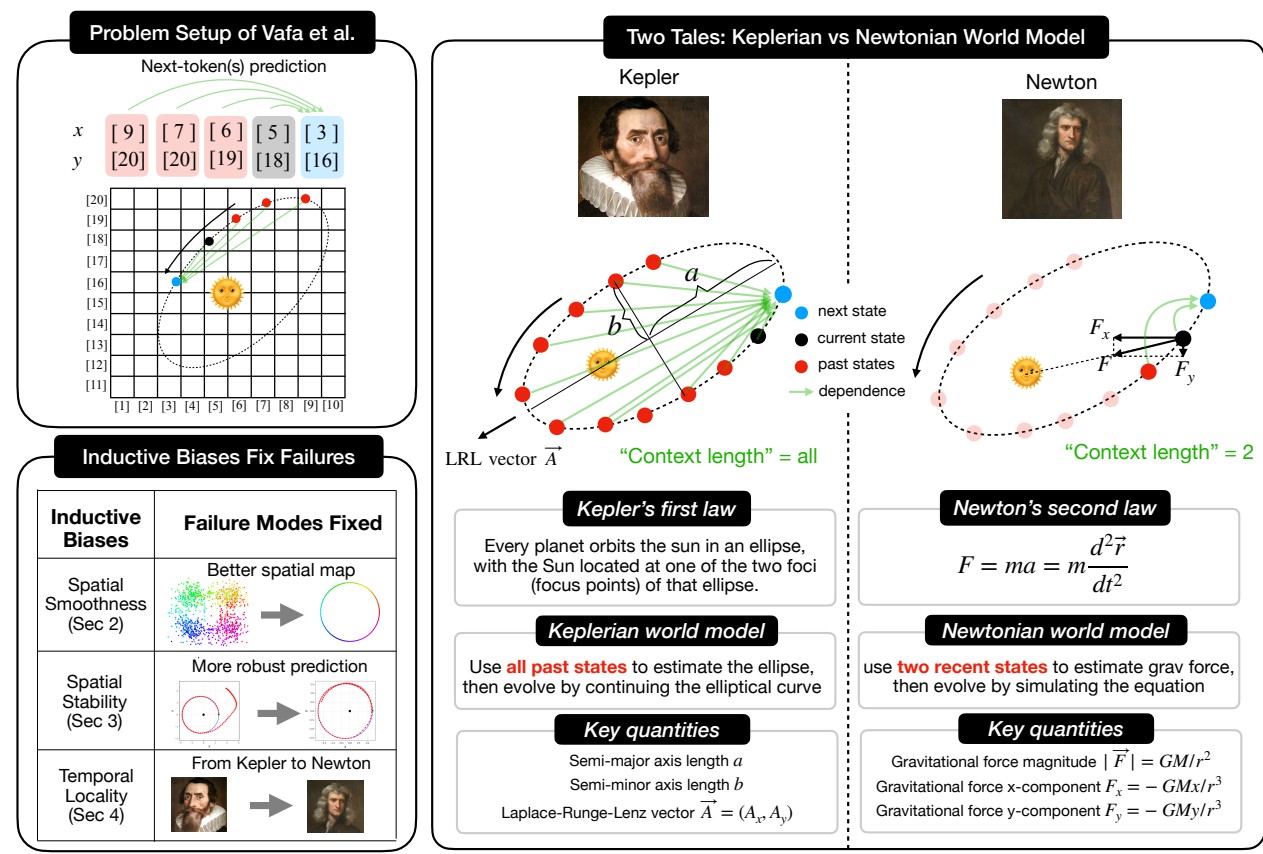

*Figure 1.* Visual abstract. Top left: The problem setup of Vafa et al. (2025): planetary motion prediction is formulated as next token(s) prediction. Bottom left: Inductive biases are key to learning Newtonian world models. Three inductive biases are identified and used to fix respective failure modes. Right: The context length controls the world model learned by transformers. Long context lengths lead to the Keplerian model (global, geometry-based), while small context lengths lead to the Newtonian model (local, force-based).

which not only make accurate predictions but also discover symbolic laws underlying the data – i.e., they successfully recover "world models" – often in settings more complex than planetary motion. The key for these AI physicist models to succeed is that they typically incorporate stronger inductive biases than transformers. We are thus motivated to study what inductive biases are lacking in transformers and how we can fix them. We find that simple and general inductive biases, like spatial smoothness, temporal continuity and temporal locality, are powerful enough to induce correct world models. The inductive biases do not need to know that much about the underlying law to be learned, but without them, it is impossible to learn.

We identify three key inductive biases required by a world model:

**Inductive bias 1: spatial smoothness.** Default tokenization discretizes continuous spatial coordinates $\vec{r} = (x, y)$ into bins (tokens), each represented by a randomly initialized, learnable embedding vector. This discretization breaks spatial smoothness, because two points that are close in

physical space but fall into different bins are treated by the transformer as completely unrelated (at least prior to training). One might hope that the model could learn a good spatial map given enough compute and data, but the spatial map does not fully emerge in the setup of Vafa et al. (2025), even though their model size, data size and training compute are comparable to GPT-2–scale models. This spatial smoothness problem may be relevant for any training paradigm involving tokenization, which motivates us to study how the emergence of a spatial map depends on key hyperparameters, i.e., vocabulary size $V$, training data size $D$, and embedding dimension $N$, which exhibit intriguing scaling behaviors detailed in Section 2.

If one insists on using tokenization, one must carefully choose $V, D, N$ to maximize spatial map emergence. Another solution, which is arguably simpler and more natural, is to use continuous coordinates without discretizing them. This, however, would lead to a stability problem stated below.

**Inductive bias 2: spatial stability.** It is known that auto-

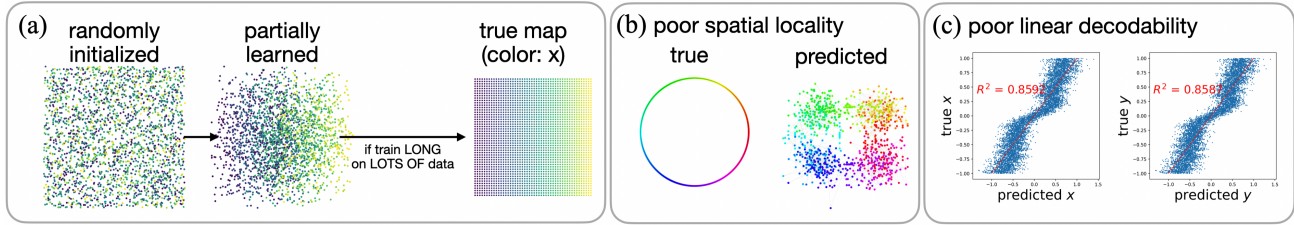

*Figure 2.* Analyzing the embeddings of the transformer model used in Vafa et al. (2025). (a) Illustration of training dynamics of token embeddings: embeddings are randomly initialized (left), gradually gain spatial structure during training (middle), requiring substantial compute and data to reach true spatial map (right). (b) The learned embeddings exhibit poor locality: circular structures in the true coordinate space (left) fragment into four point clouds, losing fine-grained structure within each quadrant (right). (c) Learned embeddings show poor linear decodability to the true spatial map (left for $x$, right for $y$).

regressive models suffer severely from error accumulation when dealing with continuous variables (Ren et al., 2025). In addition, Vafa et al. (2025) reported that discretized coordinates trained with cross-entropy loss (classification) performed better than continuous coordinates trained with MSE loss (regression). However, as continuous coordinates naturally guarantee spatial smoothness, we believe they merit further investigation. In fact, inference robustness can be significantly improved by injecting noise into the training contexts—a strategy known as noisy context learning (Ren et al., 2025). With this mitigation in place, we find that regression consistently outperforms classification across all data scales we evaluate. We elaborate on this regression-related failure mode and its remedy in Section 3.

**Inductive bias 3: temporal locality.** Newtonian mechanics has temporal locality since it is a second-order differential equation, i.e., when the time interval $\Delta t$ is small enough, the next state $\vec{r}(t + \Delta t)$ is solely dependent on the current state $\vec{r}(t)$ and the previous state $\vec{r}(t - \Delta t)$, but no states before that. This is different from a default transformer, which has a long context length (1k or longer). This inspires us to vary the context length to control temporal locality. Surprisingly, we find that: temporal locality induces the transformer to be a Newtonian world model, while lack of this knowledge induces a Keplerian world model—fitting elliptical equations based on all previous points and making predictions by continuing the curve. By contrast, a Newtonian world model would compute gravitational forces based on temporally local states and then make predictions by simulating the differential equation (see Figure 1 for an illustration). We elaborate on the two stories about Kepler versus Newton in Section 4.

The main findings and contributions in Section 2, 3 and 4 are summarized in Figure 1. Conclusions and discussions are in Section 5. Codes are available at https://github.com/KindXiaoming/newton-kepler.

## 2. Inductive Bias 1: Spatial Smoothness

### 2.1. Problem setup

Vafa et al. (2025) trained a GPT-2-scale transformer model to predict planetary motion. They reduced the problem to 2D, placing the sun at $(0, 0)$ and representing the planet's position (e.g., Earth's) in the plane as $\vec{r} = (x, y)$. The position is recorded every time interval $\Delta t$: at the $i^{\text{th}}$ snapshot (time $t = i\Delta t$), the planet's position is $(x_i, y_i)$. The transformer $f_\theta$ predicts the next position in an auto-regressive manner,

$$(x_{i+1}, y_{i+1}) = f_\theta(x_i, y_i, x_{i-1}, y_{i-1}, \ldots, x_0, y_0).$$

**Tokenization scheme.** A key design choice lies in their tokenization strategy. Rather than treating $(x, y)$ as continuous variables, $x$ and $y$ are independently discretized into bins (tokens). The procedure for $x$ is as follows (and similarly for $y$): (1) partition the interval $[-L, L]$ (with $L = 50\,\text{AU}$) into $V = 7000$ uniform bins; (2) assign each $x \in [-L, L]$ to the $k^{\text{th}}$ bin via $k = \lfloor (x/L + 1)\, V/2 \rfloor$; (3) associate the $k^{\text{th}}$ bin ($k = 0, 1, \ldots, V - 1$) with a token embedding $\vec{E}_{x,k} \in \mathbb{R}^{n_{\text{model}}}$, where $n_{\text{model}} = 768$. Likewise, $y$ coordinates use embeddings $\vec{E}_{y,k} \in \mathbb{R}^{n_{\text{model}}}$. These token embeddings are randomly initialized and learned during training. After tokenization, the continuous regression task becomes a next-token prediction (NTP) problem (Figure 1, top left). The only difference from standard language modeling is that the model outputs two tokens at each step (one for $x$, one for $y$). With this formulation, training follows the NanoGPT framework (Karpathy, 2024) using a GPT-2-scale transformer with an NTP cross-entropy loss.

**Tokenization disrupts spatial smoothness** because $\vec{E}_{x,k_1}$ and $\vec{E}_{x,k_2}$ are randomly initialized and therefore uncorrelated for $k_1 \neq k_2$, regardless of how close $k_1$ and $k_2$ are in the physical space. Although one might hope that the transformer could learn a meaningful spatial map given enough data and training time, we show that the learned embedding space does not contain a good spatial map (see Figure 2). This holds despite the fact that Vafa et al. (Vafa et al., 2025)

trained a GPT-2-scale model for days on $8\times$H100 GPUs using a massive dataset containing 20B training tokens.

## 2.2. The emergent spatial map is poor

**Linear probing** Recent work in mechanistic interpretability shows that many concepts correspond to linear directions in a model's embedding space. For example, a world map can emerge when token embeddings are projected onto a suitable 2D plane (Gurnee & Tegmark, 2024). In the same spirit, we search for linear directions in the token embedding space that correlate most strongly with the true spatial coordinates. We treat $x$ and $y$ independently. Take $x$ as an example: we aim to find a direction $\vec{t} \in \mathbb{R}^{d_{\text{model}}}$ such that $\vec{t}_x = \arg\min_{\vec{t}} \sum_i \|\vec{E}_{x,i} \cdot \vec{t} - x_i\|^2$ and we use the coefficient of determination $R^2$ to measure the goodness of fit. If $R^2 \approx 1$, then a clean linear direction corresponding to the $x$ coordinate exists. Before training, $R^2 \approx 0$ because token embeddings are randomly initialized.[1]

**Low $R^2$ indicates a poor spatial map** Using the pretrained checkpoint released by Vafa et al. (2025), we obtain $R^2 \approx 0.86$ for both $x$ and $y$. Although this is much better than random initialization – indicating that the spatial map has partially emerged – it is still far from satisfactory. The model captures coarse spatial continuity but fails to learn fine-grained locality. To illustrate this, Figure 2(c) compares the true coordinates with the predicted coordinates along the best linear directions. While a global linear trend is present, substantial deviation from the ideal $R^2 \approx 1$ remains. As a consequence, panel (b) shows that a circular orbit in real space is "perceived" by the model as a fuzzy point cloud: the global circular structure is weakly preserved (as indicated by color coding across quadrants), but local structure within each quadrant is highly distorted and noisy.

**Poor spatial map is a deal breaker for world models** Vafa et al. (2025) reported that the transformer's internal representations fail to encode the gravitational law $F \propto 1/r^2$, where $r$ is the distance between two bodies. Computing $r$ requires an accurate spatial map; without one, the model cannot recover distances reliably, let alone the gravitational force. Although the spatial map might, in principle, be stored *nonlinearly*, we show that a high-quality *linear* spatial map is achievable with appropriate hyperparameter choices.

## 2.3. Conditions and scaling laws for spatial map emergence

Since the spatial map is only weakly emergent in the setup of Vafa et al. (2025), it is important to understand the conditions under which a spatial map *does* emerge. We con-

jecture that both data coverage and model capacity play crucial roles: (1) *Data coverage:* the training data must adequately cover all tokens in the vocabulary, motivating us to vary both the training size $D$ and the vocabulary size $V$; (2) *Model complexity:* we vary the embedding dimension $N$ while keeping other hyperparameters fixed.[2]

To simplify the setting while retaining the essential features of tokenization, we adopt a 1D sine-wave dataset, which qualitatively resembles the oscillatory behavior of planetary motion but reduces the problem to one dimension.

**1D sine-wave dataset** A 1D harmonic oscillator has sine-wave solutions $x(t) = A\sin(\omega t + \varphi)$. We choose $\Delta t = 0.2$ and $T = 20$, yielding $T/\Delta t = 100$ points per trajectory. We sample $A \in U[0.5, 1]$, $\omega \in [0.5, 2]$, and $\varphi \in [0, 2\pi)$ to generate $D_{\text{traj}}$ trajectories (equivalently $D \equiv 100D_{\text{traj}}$ training tokens). Since $x \in [-1, 1]$, we partition this range uniformly into $V$ bins/tokens, converting each trajectory into a sequence of token IDs, e.g., $[6, 12, 17, 20, 21, 19, \ldots]$. Transformer models are trained using next-token prediction with cross-entropy loss. We use the Adam optimizer (Kingma, 2014) for $10^4$ steps at learning rate $10^{-3}$, followed by $10^4$ steps at learning rate $10^{-4}$. As in the previous subsection, we apply linear probing to the transformer's embedding matrix to measure whether a linear direction correlates with the true spatial coordinate, producing an $R^2$ score. We study how $R^2$ depends on three key parameters: training size $D$, vocabulary size $V$, and embedding dimension $N$. The $R^2$ we report here is the highest $R^2$ (or lowest $1 - R^2$) in training. Detailed training dynamics of $R^2$ are included in Appendix B.1.

**Larger data size $D$ improves spatial map emergence** As shown in Figure 3(b), increasing $D$ consistently improves $R^2$. This is expected: more data imposes stronger constraints on the hypothesis space, reducing overfitting and encouraging the model to learn the true spatial structure.

**Smaller vocabulary size $V$ improves emergence** Figure 3(a) shows the training dynamics of embedding projections onto the best linear direction. For a fixed amount of data ($D = 10^4$), smaller vocabulary sizes produce significantly better spatial maps. Intuitively, larger vocabularies require proportionally more data to maintain adequate coverage before a spatial map can emerge.

We fix $N = 32$ and sweep $V$ in $\{64, 128, 256, 512, 1024\}$ and $D_{\text{traj}}$ in $\{64, 128, 256, 512, 1024\}$. The scaling law in Figure 3(b, left) fits well to

$$1 - R^2 \approx AD^{-\alpha_D}V^{\alpha_V} \quad (A = 0.52, \ \alpha_D = 1.15, \ \alpha_V = 1.33) \tag{1}$$

with an excellent fit ($R^2 \approx 0.995$). Since $\alpha_V \gtrsim \alpha_D$, the training size $D$ must increase at least as fast as the vocabu-

---

[1] The embedding dimension $n_{\text{model}} = 768$ is much smaller than the vocabulary size $V = 7000$. If $n_{\text{model}} > V$, one might trivially overfit even random embeddings.

[2] We choose $n_{\text{layer}} = 2$, $n_{\text{head}} = 1$.

lary size $V$ to maintain comparable spatial-map quality.

**Embedding dimension $N$ exhibits a critical value $N_c$**
One might expect that increasing the embedding dimension $N$ would help spatial map emergence, since higher-dimensional embeddings provide more "lottery tickets" for discovering a good linear direction. However, Figure 3(b, middle) shows that $1 - R^2$ decreases with $N$ only up to a critical value $N_c \approx 8$, beyond which performance rapidly plateaus. Increasing $N$ further offers little benefit.

**Scaling up** The above analyses use small-scale hyperparameters to allow sweeping many configurations efficiently. To connect more directly to the large-scale setup of Vafa et al. (2025), which uses $V = 7000$ and $N = 768$, we repeat our experiments with these values while sweeping $D$ up to $10^8$ tokens. Figure 3(b, right) shows that scaling slows markedly at large $D$, suggesting diminishing returns once the data sufficiently cover the space. Moreover, comparing $N = 768$ and $N = 32$ (with $V = 7000$ fixed) reveals that increasing $N$ offers little improvement – and may even hinder – spatial map emergence. In contrast, reducing $V$ to 128 produces a dramatic improvement.

**Take-home message** Among all levers, reducing the vocabulary size $V$ is the most effective way to improve spatial map emergence. However, choosing $V$ too small leads to overly coarse binning, reducing the accuracy of predictions. An optimal intermediate choice of $V$ should therefore balance spatial-map quality and predictive resolution – an issue we investigate in the next section. Another natural solution would be using continuous coordinate without tokenization, but it would then face a spatial stability problem discussed in the next section.

> **Inductive Bias 1: Spatial smoothness**
>
> **Failure mode:** Vafa et al. (2025)'s transformer fails to learn a spatial map in the embedding space.
> **Solution:** (1) choose a smaller vocabulary size for tokenization, or (2) use continuous coordinates without tokenization. With (2), the spatial map is by default contained in the input.

## 3. Inductive Bias 2: Spatial Stability

Following the setup of Vafa et al. (2025), we have so far treated trajectory prediction as a classification problem by discretizing continuous spatial coordinates into discrete tokens. A natural question is whether we can instead use continuous spatial coordinates directly as inputs, thereby eliminating the need to learn a spatial map altogether. Although Vafa et al. (2025) reported that discretized coordinates with cross-entropy loss (classification) outperform continuous coordinates with MSE loss (regression), we ar-

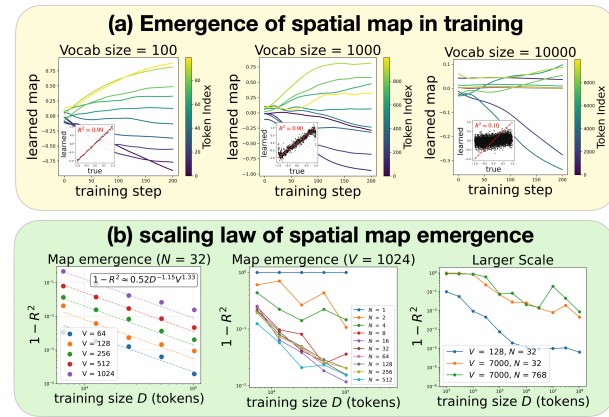

*Figure 3.* Spatial map emergence strongly depends on tokenization, and weakly on embedding dimensions. (a) Evolution of learned embeddings, i.e., token embeddings projected onto the best linearly decodable direction at the last step (200). From left to right: vocabulary size $V = 100, 1000, 10000$. Each inset shows the true coordinate and the learned coordinate. The spatial map emerges easily for a small vocabulary size $V$, but becomes poorly emergent for large $V$. (b) Spatial map quality, measured by $R^2$ between the true coordinate and the learned coordinate. Left: $1 - R^2$ obeys a scaling law with respect to vocabulary size $V$ and training tokens $D$. Middle: $R^2$ saturates when the embedding dimension $N$ is beyond a critical value at 8. Right: Scaling up $V$ or $N$ does not improve, but actually harms spatial map emergence.

gue that the continuous formulation merits further investigation. To examine this question systematically, we introduce the Kepler dataset—a simplified benchmark consisting of idealized planetary orbits. On this controlled testbed, we directly compare two formulations: next-state prediction (regression) and next-token prediction (classification).

**Kepler dataset** The Kepler dataset consists of 2D elliptical orbits of a planet around a central body (the sun) fixed at the origin. Each trajectory is generated by numerically integrating the gravitational equation of motion,

$$\frac{d^2 \vec{r}}{dt^2} = -GM \frac{\vec{r}}{\|\vec{r}\|^3}, \tag{2}$$

where $\vec{r} = (x, y)$ is the position vector and $GM = 1.0$ is the gravitational parameter. For each trajectory, we initialize the system at perihelion (closest approach) with orbital parameters sampled uniformly: eccentricity $e \in [0.0, 0.8]$, semi-major axis $a \in [0.5, 2.0]$, and initial orientation $\theta \in [0, 2\pi]$. The initial position and velocity are computed from these parameters and then rotated by $\theta$.

We integrate the system using `solve_ivp` in `scipy` with relative and absolute tolerances of $10^{-8}$, sampling 100 equally spaced time points at a step size $\Delta t = 0.2$. The resulting dataset contains $D_{\text{traj}}$ trajectories (equivalently $D = 100 D_{\text{traj}}$ tokens), each providing a sequence of $(x, y)$ positions of shape $(100, 2)$ that captures the full elliptical motion of the planet around the sun.

## 3.1. Regression: next state prediction

As discussed above, using continuous spatial coordinates as inputs naturally removes the need to learn a spatial map. However, this comes at a cost: the regression formulation is considerably more prone to error accumulation than its classification counterpart. It is well known that autoregressive models suffer from the spatial stability issue, i.e., error accumulates quickly at inference time; continuous variables exacerbate this issue because their outputs can become unbounded. In contrast, discrete tokenization imposes a finite set of possible outputs, providing a form of "error correction" by projecting predictions back onto a discrete vocabulary. Perhaps for this reason, Vafa et al. (2025) reported that "we experimented between using (a) continuous coordinates (and MSE loss) and (b) discretized coordinates (with cross-entropy loss), finding the latter worked better."

Below, we (1) identify the failure mode associated with regression and (2) introduce a mitigation strategy.

**A failure mode: error accumulation** As shown in Figure 4 (leftmost, $\sigma = 0.0$), we condition on the first 50 points of a trajectory and autoregressively generate the next 50 points, comparing them with the true trajectory. Although the initial prediction error is small, it accumulates rapidly and causes catastrophic divergence, often sending the planet into the sun or off to infinity.

**Fix: noisy context learning** To make the generation process more robust, the model must be trained to handle deviations from perfect past contexts. To simulate such deviations, we add Gaussian noise to the historical inputs during training, leading to the modified objective

$$L = \sum_{i=1}^{T} \|\vec{r}_{i+1} - f_\theta(\vec{r}_i + \sigma\vec{\epsilon}_i, \cdots, \vec{r}_0 + \sigma\vec{\epsilon}_0)\|^2, \quad (3)$$

where $\vec{\epsilon}_i \sim \mathcal{N}(0, \mathbf{I}_2)$ is drawn from a normal distribution. This is a technique known as noisy context learning (Ren et al., 2025). Figure 4 shows that an intermediate noise level ($\sigma = 0.1$) yields the most accurate predictions: too little noise fails to counteract error accumulation, while too much noise overwhelms the learning signal.

## 3.2. Fair comparison: regression wins over classification

To fairly compare regression and classification models, two subtleties must be addressed. (1) *Hyperparameter choices.* We have shown that vocabulary size $V$ is a crucial hyperparameter for classification, while the noise scale $\sigma$ strongly affects regression performance. Thus, for each method, we sweep over the relevant hyperparameters and report the performance of the best model. (2) *Evaluation metric.* Because classification models are trained with cross-entropy loss and regression models with MSE loss, their training losses are not directly comparable. Instead, we convert predicted to-

kens into continuous coordinates $(x, y)$ by mapping each token to the center of its corresponding bin. Using the first 50 points as context, we autoregressively generate the next 50 points. The evaluation metric is the *mean distance error*, defined as the Euclidean distance between generated and ground-truth positions, averaged over the 50 generated points. Detailed training dynamics are included in Appendix B.2 (classification) and B.3 (regression).

Figure 5 reports the mean distance error for both regression and classification models. In the left panel, we confirm that an intermediate noise scale $\sigma$ yields the best regression performance. In the middle panel, we observe a sweet spot for vocabulary size $V$, with the optimal $V$ increasing with the training size $D$. In the right panel, regression models (solid green) consistently outperform classification models (solid blue) across all training sizes when hyperparameters are optimized (choosing $V$ for classification and $\sigma$ for regression). However, if $\sigma$ is naively fixed to zero, regression models (dashed green) may underperform the best classification models (solid blue) at large training sizes – precisely the regime considered by Vafa et al. (2025).

> **Inductive Bias 2: Spatial Stability**
>
> **Failure mode:** Transformers (auto-regressive models in general) accumulate errors fast, especially for unbounded continuous variables.
> **Solution:** forcing the model to correct errors in inference by adding input perturbations in training. As a result, regression models with continuous inputs outperform classification models with tokenization.

# 4. Inductive Bias 3: Temporal Locality

Having established in the previous sections that regression models have two key advantages over classification models – (1) they naturally preserve spatial continuity without needing to learn a spatial map, and (2) they can achieve lower mean prediction error—we now return to the central question: *do regression transformers learn a Newtonian world model*? There is one more inductive bias needed – temporal locality. We note that Newtonian mechanics is a second-order differential equation, meaning that the next state depends only on the current state and the previous state, but not on other states before that. This suggests using a context length of 2. This motivates us to vary context length to control temporal locality.

**Newtonian world model** We apply linear probing to search for linear directions in the model's latent representations that correlate with force variables: force magnitude $F \equiv \|\vec{F}\|$, the $x$-component $F_x$, and the $y$-component $F_y$. We probe a wide range of internal activations, including the inputs and outputs of attention and MLP blocks (both before

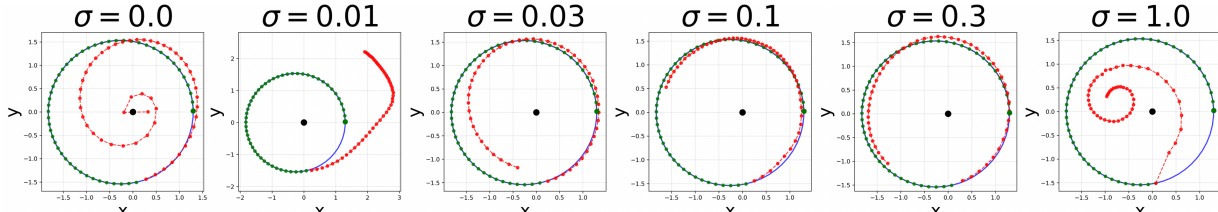

*Figure 4.* Error accumulation and fixing it by adding context noise in training. Each subplot shows the ground truth trajectory (blue solid circle), conditioning 50 points (green), and the generated 50 points (red). From left to right: training with different levels of context noise $\sigma$. Naively training a regression-based transformer leads to severe error accumulation (left, $\sigma = 0$), whereas adding a reasonable amount of noise $\sigma$ (e.g., $\sigma = 0.1$) to contexts during training substantially improves robustness.

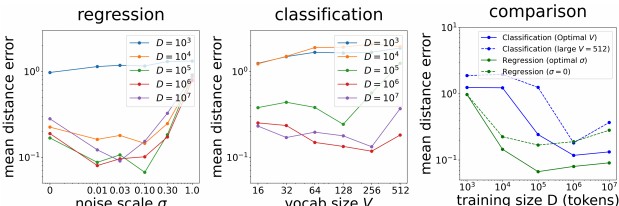

*Figure 5.* Comparing regression and classification transformers ($D$: training tokens), using mean distance error as the metric to evaluate predictive performance. Left: regression models exhibit a sweet spot in the context noise scale $\sigma$. Middle: regression models also exhibit a sweet spot in the vocabulary size $V$. Right: comparing regression and classification across different training data sizes $D$. Regression models consistently outperform classification models when their best hyperparameters ($\sigma$ or $V$) are selected. However, naively trained regression models ($\sigma = 0$) underperform the best classification models when the training data is large.

and after residual merging), as well as hidden states inside MLP modules. For each force-related target, we report the highest $R^2$ across all layers (Results for $R^2$ of each layer are included in Appendix C). Figure 6(a) (green bars) reveals that while context length 100 partially captures these force variables ($R^2 \approx 0.9$), only context length 2 yields a precise representation $R^2 \approx 0.999$.

**Keplerian world model** This raises the question: *what world model does a transformer learn when the context length is large?* It is obvious that the model must have learned something meaningful in order to make accurate predictions. If we adopt Kepler's geometric perspective – orbits are ellipses (Kepler's first law), we can first fit an elliptical equation based on all previous points and then predict the next state by continuing the curve. We refer to this global geometric approach as *Keplerian world model*, in contrast to *Newtonian world model* based on force computations (see Figure 1 right for illustrations). To test this hypothesis, we probe the model for key geometric parameters of ellipses: semi-major axis $a$, semi-minor axis $b$, Laplace–Runge–Lenz vector $\vec{A} = (A_x, A_y)$. Figure 6(b) shows that these geometric quantities are linearly encoded almost perfectly ($R^2 \approx 0.998$) when the context length is 100, whereas they are relatively poor ($R^2 \approx 0.9$) when the

context length is 2. There is a phase transition between Keplerian and Newtonian models by varying context lengths, as shown in Figure 6(c) – while transformers with *small* context lengths preferentially learn *Newtonian world models*, transformers with *large* context lengths preferentially learn *Keplerian world models*. Furthermore, Figure 6(d) shows that larger context lengths achieve lower prediction error, because the Keplerian model (global, geometry-based) is more robust to noise than the Newtonian model (local, force-based), hence making more accurate long-horizon predictions.

> **Inductive Bias 3: Temporal Locality**
>
> **Failure mode:** A regression transformer fails to learn the Newtonian model when the context length is too large.
> **Solution:** Reducing the context length to 2 induces a Newtonian model. Short context lengths induce the Newtonian model, while long context lengths induce the Keplerian model.

## 5. Conclusions and discussion

The goal of this paper is to identify failure modes that prevent current foundation models from learning "world models," using a controlled setup based on synthetic planetary motion data. We find that Vafa et al. (2025) fails to learn a spatial map due to its tokenization scheme. After addressing this issue—either by reducing the vocabulary size or by using regression models trained with MSE loss—Newton's world model still does not emerge. Instead, a geometric approach that fits ellipses (*Kepler's world model*) emerges. Newton's world model appears only when the block size is reduced to 2, consistent with the fact that Newtonian mechanics is a second-order dynamical system. Our findings underscore the challenges and subtleties involved in inducing even very simple world models (such as planetary motion) in transformer architectures.

Finally, our results offer a critical perspective on the definition of "world models" in artificial intelligence. Please refer

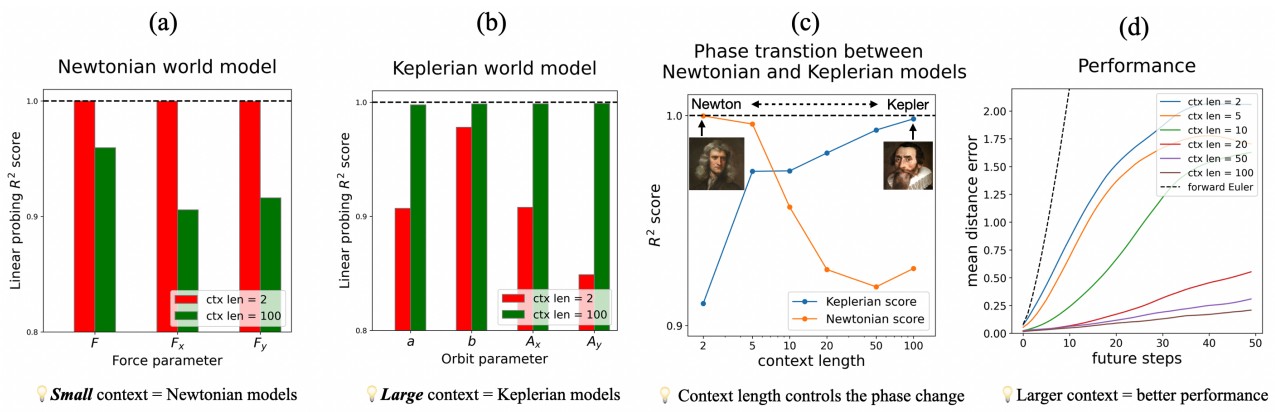

*Figure 6.* The tale of two world models – the context length controls the transformer to learn a Newtonian model or a Keplerian model. (a) Small context lengths (e.g., 2) lead to Newtonian models, with transformers internally computing gravitational forces. (b) Long context lengths (e.g., 100) lead to Keplerian models, with transformers internally computing orbit parameters (semi-major/minor-axis length $a/b$, Laplace-Runge-Lenz vector $\vec{A}$). (c) Varying context lengths controls the phase transition between Newtonian vs Keplerian world models. (d) Mean distance error as a function of future steps and context lengths. Larger context lengths lead to improved predictive performance (lower errors).

to Appendix A for a summary of previous definitions. While **prevailing views** often characterize a world model purely by its predictive utility—the ability to simulate future states given an action—we argue that prediction is **necessary but not sufficient**. As our "Keplerian" models demonstrate, a system can achieve high predictive accuracy by merely fitting complex curves to historical data, effectively memorizing the training distribution. However, true scientific understanding requires discovering the **governing mechanisms**—the simple, invariant laws (like $F = ma$) that generate the data.

We speculate that this mechanistic understanding is the prerequisite for **radical out-of-distribution (OOD) generalization**. A curve-fitter (Kepler) is bound to the "grey elephants" (familiar trajectories) it has seen; it cannot reliably predict the **behavior** of a "pink elephant"—a scenario strictly outside its training data. In contrast, a mechanistic model (Newton) decouples the rule from the history. Because it understands the causal invariant ($F = ma$), it can correctly predict the future of a pink elephant just as accurately as a grey one. Our work suggests that for AI to evolve from a predictor to a scientist, we must architecturally constrain it to look for these simple, local mechanisms rather than complex global histories.

**Limitations** (1) For simplicity, we have simplified the setup of Vafa et al. (Vafa et al., 2025) by using one time-scale (they used two time-scales). (2) We used linear probes to verify the existence of world models, but this method yields only implicit knowledge: the network "knows" Force, but does not explicitly output Newton's equations. Furthermore, probing requires us to know what to look for a priori. A fully autonomous "AI Physicist" would require an additional mechanism—such as a secondary "interpreter" network or

symbolic regression head—that actively searches the latent space for simple, linear relationships to automatically extract and output symbolic laws like $F = ma$ without human supervision. Related works about "AI Physicists" are reviewed in Appendix A – although they have proven useful in highly-controlled setups where carefully designed specialist models are employed, end-to-end extraction of clean and simple laws (if they do exist) from black-box foundation laws remains a challenging but urgent direction.

## Acknowledgement

We would like to thank Liam Storan for the helpful discussion. S.G thanks the Simons Collaboration on the Physics of Learning and Neural Computation and a Schmidt Sciences Polymath award for funding. Z.L, S.S and A.T thank The James Fickel Enigma Project Fund.

## Impact Statement

This work contributes to the scientific understanding of how foundation models acquire internal world models. We show that simple inductive biases can strongly influence whether a model learns a faithful representation of underlying physical mechanisms or relies on alternative predictive strategies. These findings may support the development of more interpretable, robust, and scientifically useful AI systems. Because our work is fundamental research conducted on synthetic datasets, we do not anticipate direct negative societal impacts. However, improved world-modeling capabilities may ultimately contribute to more capable AI systems, highlighting the importance of continued research on interpretability, transparency, and AI safety.

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

## A. Related works

**World models** The notion of "world models" has appeared across several threads in machine learning, generally referring to internal representations/computations that capture the dynamics or structure of an environment. Early work in model-based reinforcement learning developed predictive latent-state models for planning and control, such as the World Models framework of Ha and Schmidhuber (Ha & Schmidhuber, 2018), PlaNet (Hafner et al., 2019b), and Dreamer (Hafner et al., 2019a), which learn compact dynamic models from pixel observations. A parallel line of work in neuroscience-inspired ML explores generative predictive coding frameworks and latent dynamics models as computational analogues of internal world representations (Rao & Ballard, 1999; Friston, 2010; Lotter et al., 2017). More recently, the emergence of world models in large foundation models has attracted growing attention: language models have been shown to encode linearizable geometric or semantic structures (Gurnee & Tegmark, 2024; Park et al., 2024; Korchinski et al., 2025), vision-language models capture rich conceptual relationships (Radford et al., 2021; Goh, 2021; Alayrac et al., 2022), and multimodal agents acquire implicit affordances for acting in embodied environments (Zitkovich et al., 2023; Reed et al., 2022; Kim et al., 2024).

**AI physicists** Within physics and scientific modeling, AI physicist approaches have demonstrated that symbolic laws and interpretable dynamical equations can emerge from neural network training (Wu & Tegmark, 2019; Brunton et al., 2016; Cranmer et al., 2020; Lemos et al., 2023; Liu & Tegmark, 2021; Liu et al., 2022; 2024; Udrescu & Tegmark, 2020), suggesting that data-driven models can recover ground-truth world structures under the right inductive biases. Crucially, these methods typically impose strong structural priors (e.g., sparsity, symmetry, or graph structure). This leaves open the question of whether general-purpose architectures like Transformers can recover such "world models" without these domain-specific constraints. Recent work by Vafa et al. (Vafa et al., 2025) suggests they cannot, reporting negative results even on simple planetary-motion datasets. Our work builds on these lines of inquiry but demonstrates that recovering true physical laws does not require strong symbolic priors, but rather a minimal assumption of locality—the intuitive idea that nature is governed by simple local rules rather than a complex history of the past. By constraining the model to seek such simple explanations, we show that general-purpose architectures can recover the ground-truth world structure without domain-specific constraints.

## B. Training dynamics

### B.1. 1D sine wave dataset

In the main paper, for the 1D sine-wave dataset, we report the best (lowest) value of $1 - R^2$ attained during training. This choice is necessary because the optimal mapping is not always achieved at the final training step: the training dynamics can exhibit non-monotonic behavior due to overfitting. Figure 7 illustrates the evolution of the training loss, test loss, and map quality $R^2$ for different vocabulary sizes and numbers of training tokens. For small training budgets (e.g., $10^4$ tokens), severe overfitting is observed, as evidenced by a large gap between the training and test losses. In this regime, both the test loss and the map quality evolve non-monotonically: they initially improve but subsequently degrade once overfitting sets in. Notably, with limited data, a smaller vocabulary size ($V = 128$) yields better maps than a larger one ($V = 7000$), highlighting the advantage of smaller vocabularies in the low-data regime. Larger vocabulary sizes require more data to avoid overfitting. For example, with $10^6$ training tokens (middle column), the train–test gap is negligible for $V = 128$ but remains noticeable for $V = 7000$. However, larger vocabularies benefit from longer scaling as the training data increases: for $V = 7000$, performance continues to improve when scaling from $10^6$ to $10^8$ tokens, whereas such gains largely saturate for $V = 128$.

### B.2. Kepler as classification

When we formulate the Kepler problem as a classification task (next-token prediction) trained with cross-entropy loss, two key hyperparameters are the vocabulary size and the number of training tokens. As shown in Figure 8, overfitting can occur when the training data are limited, especially for large vocabulary sizes. Although the model is trained using cross-entropy loss, we additionally compute an effective MSE loss defined by the squared distance between the center of the predicted token and the true next position. This allows for a fair comparison with the regression results shown in Figure 9.

### B.3. Kepler as regression

When we formulate the Kepler problem as a regression task (next-state prediction) trained using MSE loss, two key hyperparameters are the in-context noise level and the number of training tokens. As shown in Figure 9, overfitting can

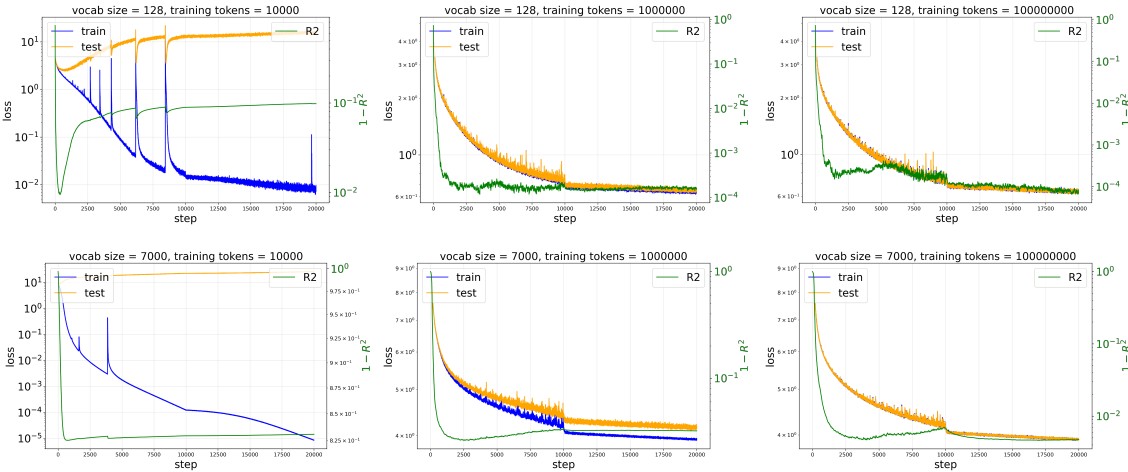

*Figure 7.* Training dynamics of the training loss, test loss, and $R^2$ for the 1D sine-wave dataset, with vocabulary sizes 128, 7000 and training token counts $10^4, 10^6, 10^8$. When the training set is small, the learned map initially improves (lower $1 - R^2$) but subsequently degrades as overfitting sets in. Larger vocabulary sizes require more data to mitigate overfitting, as reflected by a larger train–test loss gap.

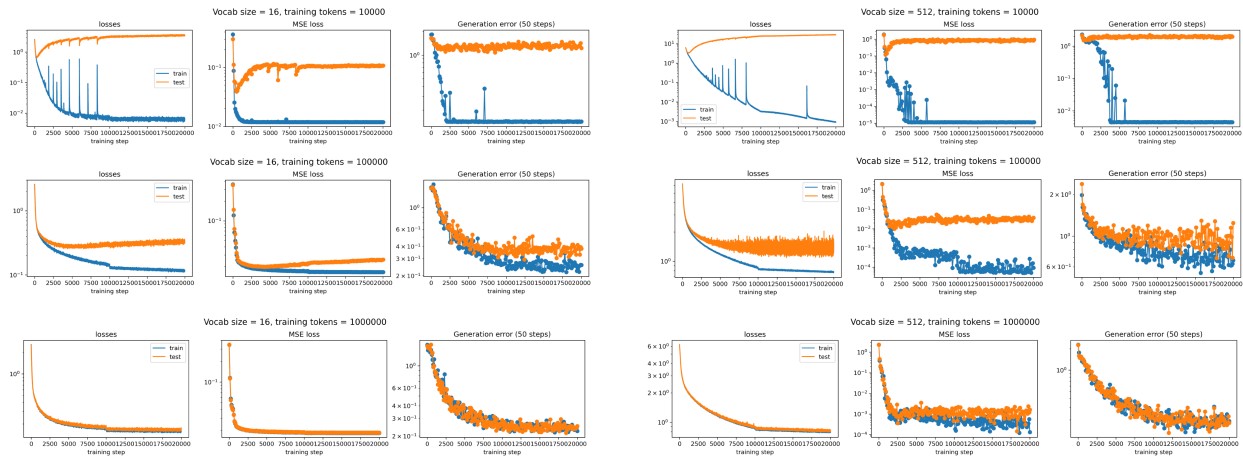

*Figure 8.* When formulating the Kepler problem as a classification task (next-token prediction), performance depends on both the vocabulary size and the number of training tokens. For each configuration, we plot the training dynamics of the cross-entropy loss, the effective MSE loss (defined as the squared distance between the true point and the center of the predicted token), and the generation distance error averaged over the next 50 steps. When the training data are limited, smaller vocabulary sizes yield better performance.

occur when the training data are limited; however, the amount of data required to avoid overfitting is smaller than in the classification setting shown in Figure 8. For example, with $10^5$ training tokens, a noticeable train–test gap remains for classification, whereas it is negligible for regression. In addition, introducing a moderate level of in-context noise during training ($\sigma$) helps reduce generation error, as demonstrated in Figure 4.

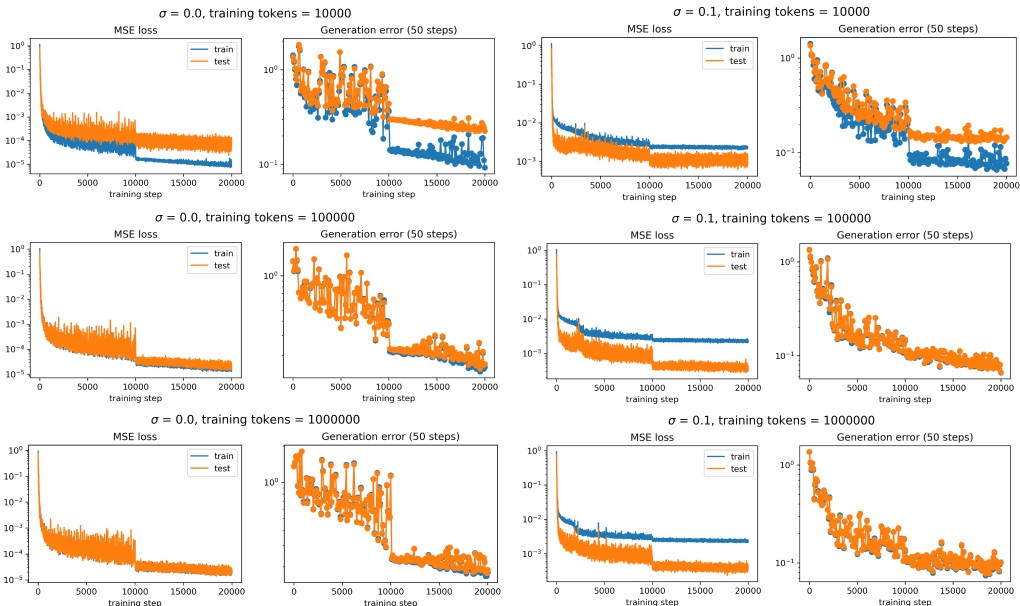

*Figure 9.* When formulating the Kepler problem as a regression task (next-state prediction), performance depends on both the noise scale and the number of training tokens. For each configuration, we plot the training dynamics of the MSE loss and the generation distance error averaged over the next 50 steps. Introducing a moderate level of in-context noise during training offers advantages over training without noise.

## C. More probing results

### C.1. Interpolating between Kepler and Newton by varying context lengths

In the main paper, for the Kepler model, we showed that a transformer with a short context length (2) learns a Newtonian model, whereas a transformer with a long context length (100) learns a Keplerian model. What happens when we interpolate between these two extremes? Figure 10 shows that the transition is monotonic: increasing the context length yields models that are progressively more Keplerian and less Newtonian. For simplicity, the metrics in the rightmost plot are averaged across components (i.e., the Keplerian score averages over $a, b, A_x, A_y$, while the Newtonian score averages over $F, F_x, F_y$).

### C.2. Probing results across layers

In the main paper, we report the best $R^2$ over all hidden representations. But where are the best representations, and how do they emerge in forward computations? For context length 2, probing results are shown in Figure 11. For context length 100, probing results are shown in Figure 12. Besides the probing targets reported in the main paper, we compute other related quantities as well.

**Keplerian model**: semi-major axis length $a$ (and $1/a$, $1/a^2$), semi-minor axis length $b$ (and $1/b$, $1/b^2$), half focal length $c = \sqrt{a^2 - b^2}$, ellipticity $e = c/a$, average radius $\bar{r} = \sqrt{ab}$, Laplace-Runge-Lenz vector $\vec{A} = (A_x, A_y)$ and magnitude $|\vec{A}|$, radial unit vector $\hat{r} = (n_x, n_y)$.

**Newtonian model** Gravitational force $\vec{F} = (F_x, F_y)$ and magnitude $|\vec{F}|$, force direction $(\hat{F}_x, \hat{F}_y) = \hat{r} = (n_x, n_y)$, position $\vec{r} = (x, y)$, radius $r = \sqrt{x^2 + y^2}$ (and $r^2, 1/r, 1/r^2, 1/r^3$).

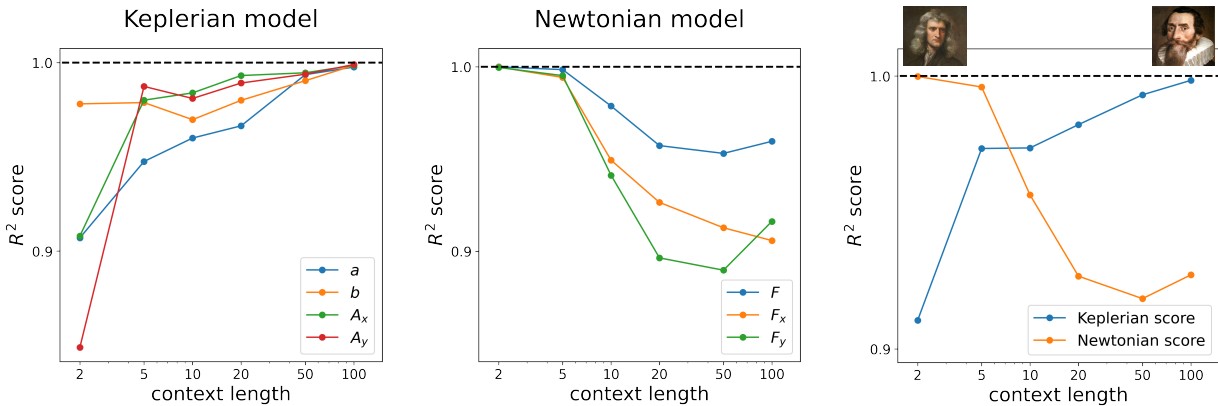

*Figure 10.* Effect of context length on the learned world model. Left: Larger context lengths favor the emergence of a Newtonian world model. Middle: Smaller context lengths favor the emergence of a Keplerian world model. Right: A summary plot obtained by averaging the $R^2$ scores across different probing targets.

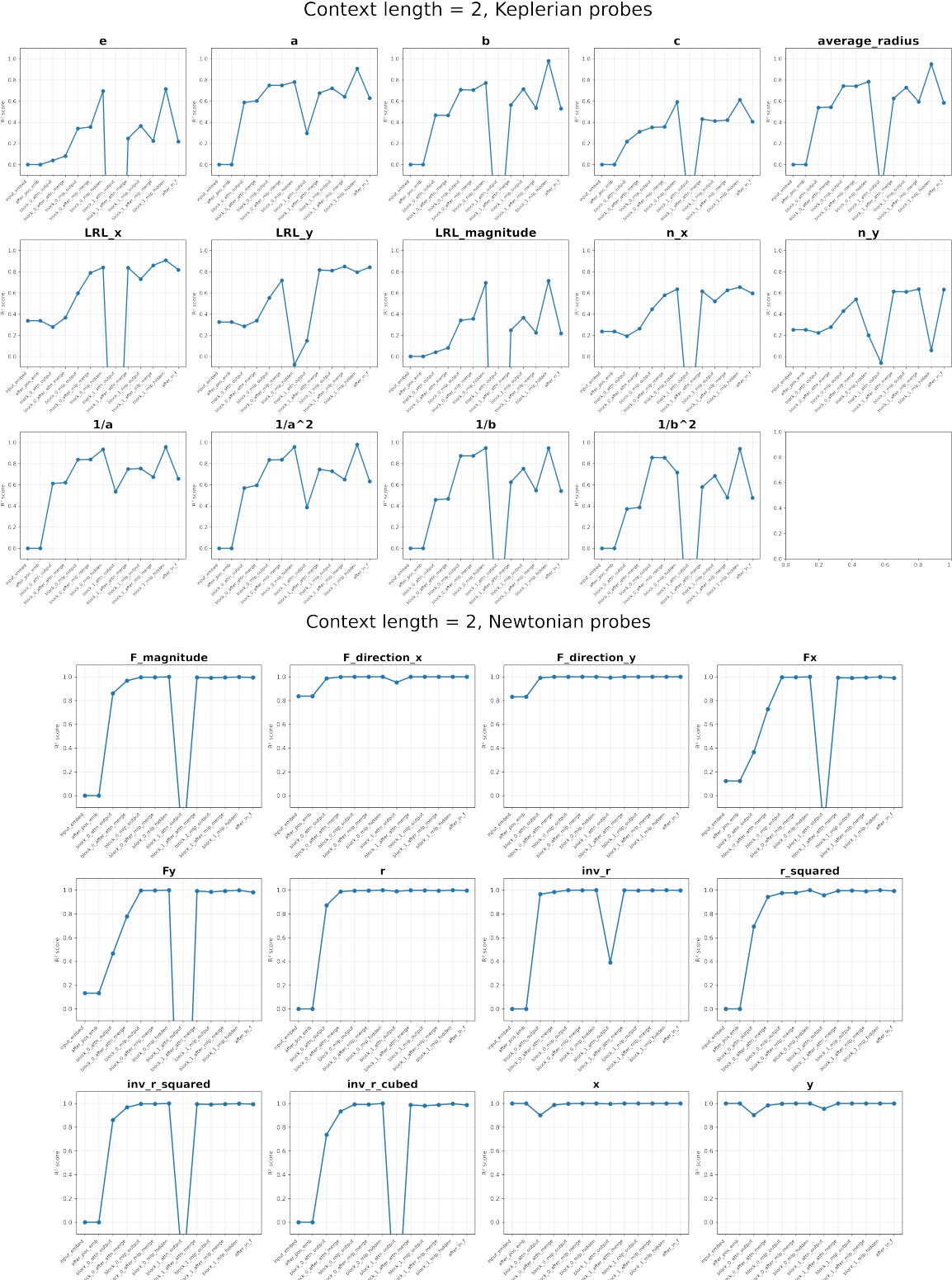

*Figure 11.* Probing results for context length 2. Top: Probes related to the Keplerian model. Bottom: Probes related to the Newtonian model.

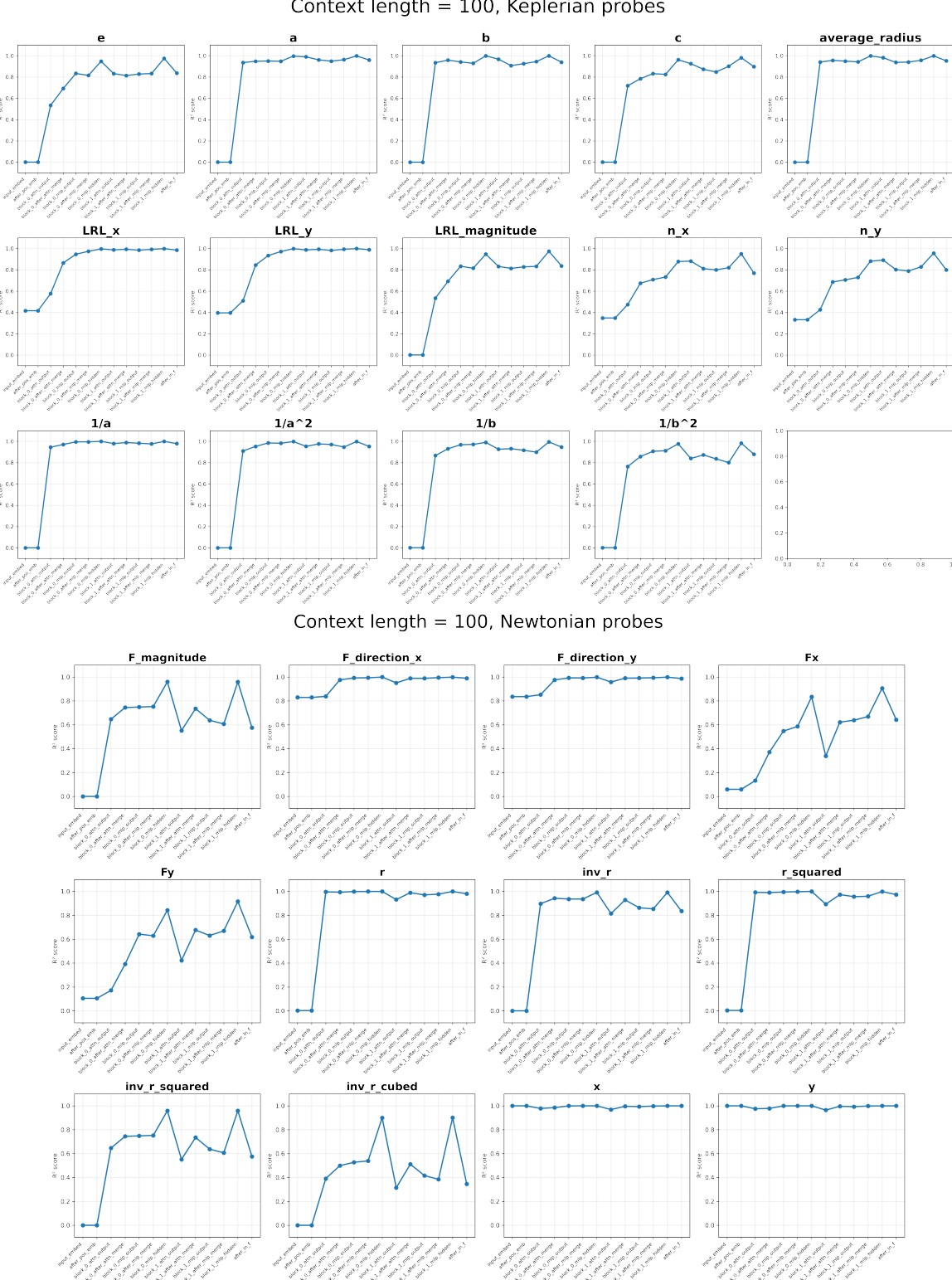

*Figure 12.* Probing results for context length 100. Top: Probes related to the Keplerian model. Bottom: Probes related to the Newtonian model.

