# OpenReview forum: "From Kepler to Newton: Inductive Biases Guide Learned World Models in Transformers"
_ICML.cc/2026/Conference — ICML 2026 regular_

### Official Review · Reviewer_vbaT · 2026-02-24

**Soundness:** 2
**Presentation:** 3
**Significance:** 3
**Originality:** 3
**Overall Recommendation:** 4
**Confidence:** 4

**Summary:**

This paper discusses why transformers may fail to recover the underlying Newtonian rules and instead tend to learn a Keplerian representation. It shows that adding three key inductive biases can help transformers recover Newtonian rules in this controlled setting.

**Compliance With Llm Reviewing Policy:**

Affirmed.

**Final Justification:**

This paper explores the inductive bias of transformers in scientific discovery. It addresses an interesting topic and has the potential to make an impact on the community.

**Key Questions For Authors:**

1. If sufficiently strong inductive biases are introduced, it seems possible that a much simpler model could already be sufficient. I mean, some simple methods should also be used as ablation studies.
1. The paper should define “scientific discovery” more clearly, and earlier. Is the goal a form of fitting with inductive biases, or actual formula/law discovery? The paper discusses this to some extent later, but it would be much better to state a clear definition earlier. I understand that this may be inherently difficult to define, but at least provide a definition within the framework of this paper.
1. How general are these inductive biases? For example, in domains such as quantum chemistry, would a different set of assumptions be required? From this perspective, the proposed inductive biases can feel somewhat like injecting the answer structure and then recovering the expected result.
1. What happens if the true trajectory is spiral-shaped? Can the model still predict correctly in that case?

If the questions can be well addressed, I'll raise the score.

**Limitations:**

yes.

**Strengths And Weaknesses:**

- Soundness: The paper uses a highly controlled experiment to uncover failure modes, which is a reasonable approach. However, it lacks evidence that the conclusions can be generalized beyond this setup. This weakens the paper’s soundness.

- Presentation: The figures are intuitive and the descriptions are clear. Using Kepler and Newton to illustrate the two paradigms is both interesting and very intuitive. There are some typos, e.g., ??.

- Significance: The paper investigates an important issue in current Transformer-based research. At present, many studies use Transformers without sufficient reflection, which is an incorrect practice. This paper highlights the importance of inductive bias, which can provide useful support for future research and therefore has substantial value.

- Originality: Although the methods themselves are not new, the paper provides useful insights, which is a strong point. However, the generalizability of these insights is not sufficiently studied or justified.

---

> ### Author Rebuttal · Authors · 2026-03-30
>
> We would like to thank the reviewer for their kind words and constructive suggestions. Below we respond to the weakness points and questions.
> * A much simpler model could already be sufficient.
>     * Agreed! For example, Neural ordinary differential equations (Neural ODEs) are good models for Newtonian laws. We still want to keep the transformer architecture though because the very point of this paper is to analyse emergence of world models on transformers.
> * This paper should define “scientific discovery” more clearly
>     * This is a good point! Although the ultimate dream is to define “scientific discovery” more broadly, this paper defines it in a narrow sense — a model is said to discover a physical variable if its hidden representations linearly encode the value of the physical quantity.
> * How general are these inductive biases? For other domains such as quantum chemistry, a different set of assumptions may be required.
>     * It might be hard to quantify the generality of inductive biases, but many scientists believe that natural systems at least obey some extent of spatial & temporal locality, otherwise the system might be too chaotic or unstable to exist in our universe.
> * What happens if the true trajectory is spiral-shaped? Can the model still predict correctly in that case?
>     * We expect the “Keplerian model” to be applicable, as long as the trajectory has simple parametric forms. For example, an ellipse is parametrised by major/minor axis length, and a spiral also has parametric forms (something like $r = a e^{b \theta}$, in polar coordinates). So we expect a Keplerian model (with long contexts) would learn a spiral as well. A Newtonian model (with context length = 2) could also learn a spiral by adding dissipation forces in addition to Newtonian gravity.

---

> > ### Author Rebuttal · Reviewer_vbaT · 2026-04-03
> >
> > Thanks for the authors’ response. I still have some follow-up questions, but I think they may be addressable.
> > Q2 & Q3. Inductive biases are inherently conditional. For example, if human reasoning strictly constraints to classical assumptions like continuity, we might never have reached quantum theory. I suggest the authors explicitly discuss this.

---

### Official Review · Reviewer_MRBY · 2026-03-12

**Soundness:** 3
**Presentation:** 3
**Significance:** 3
**Originality:** 3
**Overall Recommendation:** 5
**Confidence:** 4

**Summary:**

In this paper, the authors revisit why generic transformers can predict planetary motion without internalizing Newtonian physics, and argue that three missing inductive biases, spatial smoothness, spatial stability, and temporal locality, explain the gap: tokenization destroys local geometry, continuous autoregression suffers error accumulation, and long contexts encourage the model to rely on global history rather than local laws. They show that spatial structure improves mainly by reducing vocabulary size or by switching to continuous coordinates, and on a simplified Kepler dataset they find that regression with noisy-context training fixes instability and outperforms tokenized classification. However, these fixes alone still tend to produce a “Keplerian” model that extrapolates ellipses from long histories; only when context length is reduced to 2, matching the locality of a second-order dynamical system, does the transformer more cleanly encode Newtonian force variables, even though the longer-context Keplerian model can predict more accurately over long horizons. The paper’s broader conclusion is that predictive accuracy is not the same as scientific understanding: a model can forecast well by curve-fitting, but recovering mechanistic, causal laws requires architectural pressure toward simple local rules, which the authors present as a step toward more genuine machine “world models.”

**Compliance With Llm Reviewing Policy:**

Affirmed.

**Final Justification:**

I have changed my score based on the rebuttal.

**Key Questions For Authors:**

N/A

**Limitations:**

yes

**Strengths And Weaknesses:**

weaknesses:
1- The central “Newtonian vs. Keplerian world model” claim is not cleanly identified from the evidence.
The paper’s main evidence in Section 4 is probe-based: it searches many internal activations and reports the best R^2 across layers, then interprets high probe scores as evidence that a short-context model is “Newtonian” and a long-context model is “Keplerian.” But the separation is soft, not clean: the long-context model still has force probes around R^2=0.9, while the short-context model still has geometric/orbit probes around R^2=0.9. Appendix C.2 further shows that they probe a large family of tightly related variables, position, radius, force direction, orbital elements, etc., so in this analytically simple Kepler problem, many targets are mathematically entangled. That makes it hard to infer a unique internal algorithm from decodability alone. The authors themselves partly acknowledge this in the limitations: probes yield only implicit knowledge and require knowing what to look for a priori. So the strongest supported conclusion is “probe prominence shifts with context length,” not necessarily “the model has become a physicist rather than a curve-fitter.”

2- The positive result is heavily task-engineered, so the claimed inductive biases may be much less general than advertised.
Their benchmark is a very favorable case: idealized 2D ellipses around a fixed sun, with sampled orbital elements, and they admit in the limitations that they simplified the original setup to a single time-scale. More importantly, the decisive intervention in Section 4 is to set context length to 2 specifically because Newtonian mechanics is a second-order system; in the conclusion they say Newton’s world model appears only when block size is reduced to 2. That is not a weak, generic bias, it is very close to giving the model the correct causal order of the true law. Section 2 is also partly built on a proxy task: the spatial-smoothness/scaling story is studied on a 1D sine-wave toy rather than the full orbital system. This matters because the paper’s big message is that “simple and general” inductive biases determine whether a transformer becomes a physicist, but the evidence may instead show something narrower: if you simplify the physics enough and insert a near-correct locality prior, you can coax the desired representation out.

3- The paper’s strongest interpretive and impact claims are largely untested.
In Section 5, the paper argues that high-accuracy “Keplerian” models are effectively memorizing the training distribution and then says, explicitly, “We speculate” that mechanistic understanding is the prerequisite for radical OOD generalization. But that OOD claim is not actually evaluated. Likewise, the authors admit that their probe evidence is only implicit, the network “knows” force, but does not explicitly output Newton’s equations, and that an additional interpreter or symbolic-regression mechanism would be needed to extract laws like
F=ma. That gap matters because the paper frames itself as moving AI from “predictor to scientist,” yet the demonstrated result is much narrower: better forecasting plus probe-decodable latent variables in a toy domain. So the paper’s impact story may be directionally plausible, but it is not yet empirically established by the presented experiments.


Strengths:
1- It turns a vague negative result into a crisp explanatory framework.
The strongest conceptual contribution is not any single trick, but the paper’s decomposition of transformer failure into three minimal inductive biases, spatial smoothness, spatial stability, and temporal locality, and its claim that each fixes a distinct failure mode. The abstract frames the whole contribution this way, and Figure 1 explicitly summarizes the mapping from bias to repaired failure mode and from “Kepler” to “Newton.” That matters because it upgrades the discussion from “transformers failed on this benchmark” to a reusable causal account of why they fail and what kind of extra structure is sufficient to make mechanistic world-model learning possible, without resorting to heavy domain-specific priors. This is the paper’s core conceptual bridge between generic transformers and “AI physicist” style systems.

2- Its methodology is unusually careful in isolating confounds and producing actionable design laws.
Sections 2 and 3 are strong because they do more than propose improvements; they measure the failure modes and compare alternatives fairly. In Section 2, the paper derives a scaling relation for spatial-map emergence, shows that vocabulary size is the dominant lever, and finds that embedding dimension has an early critical point and then saturates. That converts a loose intuition (“tokenization breaks continuity”) into quantitative guidance about when tokenized transformers should be expected to recover spatial structure. Then Section 3 builds a controlled Kepler benchmark, identifies the specific regression failure mode as autoregressive error accumulation, introduces noisy-context training as a targeted fix, and compares regression with classification under matched hyperparameter sweeps and a shared continuous metric (mean distance error). Under that fairer comparison, regression with noisy-context learning beats classification. This is a deep strength because it makes the paper falsifiable and practically useful, rather than just interpretive: it gives concrete rules for architecture and training choices, and it overturns the simplistic conclusion that discretization is inherently better for this problem.

3- It makes a genuinely important distinction between prediction and understanding.
The deepest empirical/theoretical insight is the paper’s “two world models” result: with short context, the transformer preferentially represents Newtonian quantities such as force; with long context, it preferentially represents Keplerian geometric quantities such as ellipse parameters. Figure 6 then adds the crucial twist: the long-context, more Keplerian models can actually have better predictive performance. Appendix C.1 strengthens the story by showing a monotonic transition as context length increases. This matters a lot for interpretation and impact. It means the paper is not just showing that one setting predicts better than another; it is showing that high predictive accuracy can come from the wrong internal scientific story. That is a powerful challenge to any definition of “world model” based only on forecasting quality. The conclusion makes this explicit by arguing that prediction is necessary but not sufficient, because a model can fit trajectories well without recovering the governing mechanism.

---

> ### Author Rebuttal · Authors · 2026-03-30
>
> We would like to thank the reviewer for their kind words and constructive suggestions. Below we respond to the weakness points and questions.
> * Probe-based results cannot cleanly separate the two mechanisms.
>     * Agreed! We will also tone down the argument to emphasise that our conclusions are solely based on probing, but not any circuit-level causal analysis.
> * The positive result is task-engineered and may not be that general.
>     * We agree that the positive result is somewhat task-engineered, but want to emphasise that it is non-trivial that a positive result can be engineered at all, given the negative results in Vafa et al. Whether these simple tricks are general enough for other world models require testing on larger-scale world models, e.g., video generative models, which will be left for future work.
> * The OOD claim is not verified.
>     * Thanks for the good point! We will make sure to tone it down. We do not have evidence for out-of-distribution generalization yet. Learning the correct equation for in-distribution data is a good sign of generalization, but does not guarantee that. The key gaps remain: how to extract the correct equation symbolically (beyond linear probing) and apply the symbolic equation to out-of-distribution data? A subtle point is to distinguish between generalization loss (due to distribution shift in orbit parameters) and robustness loss (due to error accumulation along contexts), and our results show that the Newtonian model has worse robustness loss.

---

> > ### Author Rebuttal · Reviewer_MRBY · 2026-04-02
> >
> > Authors adequately answered my comments.

---

### Official Review · Reviewer_skMG · 2026-03-12

**Soundness:** 3
**Presentation:** 3
**Significance:** 3
**Originality:** 3
**Overall Recommendation:** 4
**Confidence:** 5

**Summary:**

This paper investigates what inductive biases are needed for transformers to learn true world models: causal abstractions that capture underlying physical laws, not just predictive accuracy. The authors draw an analogy between Kepler's geometric description of planetary motion and Newton's mechanistic explanation. Building on the finding of Vafa et al. (2025) that generic transformers fail to learn a Newtonian world model for planetary motion despite high predictive accuracy, the authors systematically identify three minimal inductive biases that fix this failure. 1) Spatial smoothness: Default tokenization breaks spatial continuity, because nearby coordinates mapped to different bins receive unrelated embeddings. This prevents the model from learning a reliable spatial map, which is necessary to recover distance-dependent laws like gravity. The authors show that either using a small vocabulary size or switching to continuous coordinates (removing tokenization entirely) resolves this issue. 2) Spatial stability: Autoregressive models with continuous outputs suffer from rapid error accumulation at inference time. The authors address this via noisy context learning which forces the model to correct for imperfect past predictions. With this fix, regression models consistently outperform classification (tokenized) models across all data scales. 3) Temporal locality: When the transformer uses a long context window, it learns a Keplerian world model. Only by restricting the context length to 2 (matching the second-order nature of Newtonian mechanics) does a Newtonian world model emerge, with gravitational force variables linearly encoded in the model's representations.

**Compliance With Llm Reviewing Policy:**

Affirmed.

**Final Justification:**

After reading the other reviews and the rebuttal, I agree that this is a good work.

**Key Questions For Authors:**

You explicitly set the context length to 2 because Newtonian mechanics is a second-order differential equation. However, one primary objective of an "AI Scientist" is to autonomously discover novel physical laws and explanations without requiring a priori knowledge of their mathematical structure. Doesn't this approach contradict the vision of automated scientific discovery? Couldn't this requirement for domain-specific architectural choices severely limit the model's ability to discover genuinely novel physical laws whose mathematical structure we don't yet know?

While I appreciate that you acknowledge this limitation, could you elaborate what in this framework could be extended to scenarios where we don't know the governing equations in advance? I would like to better understand how this work, which is restricted to planetary motion prediction, can be used by someone is the field to learn better world models without handcrafting the solution within the model architecture.

**Limitations:**

yes

**Strengths And Weaknesses:**

Strengths
- Kepler vs. Newton provides an elegant and intuitive framework for understanding the distinction between curve-fitting and mechanistic world models.
- The authors perform rigorous quantitative experiments.
- Clear, actionable guidance is provided on hyperparameter choices (vocabulary size, context length, noise scale) for practitioners.
- The paper is well-written, with logical flow and clear section organization.
- Figure 1 serves as an excellent visual abstract that effectively summarizes the main problem setup.

Weaknesses
- Limited conceptual novelty: While the specific findings are interesting, the underlying challenges addressed—overfitting due to model complexity, using dimensionality reduction for generalization, and applying noise perturbations for robustness—are well-established techniques in machine learning that have been used for decades. The novelty lies primarily in applying these known solutions to the specific problem of world model learning planetary motions in transformers.
- Figure readability issues: Figure 3 suffers from poor readability due to excessively small font sizes on axis labels.
- Missing references: There is a missing section number reference (line 147: "Related works are discussed in Section ??, with conclusions...").

---

> ### Author Rebuttal · Authors · 2026-03-30
>
> We would like to thank the reviewer for their kind words and constructive suggestions. Below we respond to the weakness points and questions.
> * The novelty lies primarily in applying these known solutions to the specific problem of world model learning planetary motions in transformers.
>     * Agreed! And we may view it as an advantage — world models (of planetary motion) can be induced with well-known solutions. Of course, we still need to do more experiments to verify the generality of these existing solutions for more complicated tasks.
> * Figure readability issues
>     * We will make sure to enlarge font sizes to make subplots of Figure 3 more readable.
> * Missing section number reference
>     * We will fix this in the revised version
> * How can the framework be extended to scenarios where we don’t know the governing equations?
>     * This is a very good question. We could try training models of various complexity (e.g., context length can be viewed as a type of complexity) and select the model that strikes the best balance between simplicity and performance.
>     * The question is hard in general, because there is usually a trade-off between generality and efficiency for AI models. When the hypothesis space is large, it is general but hard to discover the equation efficiently; when the hypothesis space is small, it may discover specific equation efficiently but may not be general enough.

---

> > ### Author Rebuttal · Reviewer_skMG · 2026-04-02
> >
> > Thank you for the rebuttal, my concerns are resolved.

---

### Official Review · Reviewer_5gex · 2026-03-15

**Soundness:** 4
**Presentation:** 4
**Significance:** 3
**Originality:** 3
**Overall Recommendation:** 5
**Confidence:** 3

**Summary:**

This paper investigates why transformers trained for planetary motion prediction fail to learn Newtonian mechanics and proposes three inductive biases to fix this. 1) Spatial smoothness: tokenization of continuous coordinates breaks spatial locality and one can fix this by reducing the vocabulary size or switch to continuous regression, 2) spatial stability: continuous regression suffers from compounding errors, using noisy context learning can mitigate this, and 3) temporal locality: reducing the context length to 2 forces the model to learn Netwonian mechanics, while long context lengths let the model learn Keplerian representations. There is a clear phase transition between the two world models as context length varies. This paper is part of a broader push to make general purpose AI architectures understand fundamental underlying dynamics as world models.

**Compliance With Llm Reviewing Policy:**

Affirmed.

**Final Justification:**

The authors partially addressed my concerns and my evaluation does not change. The paper deals with the interesting problem of inductive biases and world models and does a systematic evaluation. It's very interesting and nice work.

**Key Questions For Authors:**

Questions:
- There seems to be a quite sharp transition from Newtonian to Keplerian: even context length 5 seems to be partially Keplerian already. Could you comment as to why?

**Limitations:**

The authors clearly list the limitations in great detail.

**Strengths And Weaknesses:**

Strengths:
- The research questions are very clear and appropriate. The Kepler vs. Newton framing is easily understandable. The paper is well structured, first identifying failure modes and proposing solutions, before moving to the next inductive bias.
- I think the strongest result of this paper is the clear Kepler-Newton phase transition as context length varies. This transition in Figure 6c is strikingly clear and is a great finding.
- The spatial map emergence results highlight a perhaps counterintuitive result: that V matters more than the embedding dimension. The authors also derive scaling laws, with R^2 around 0.995.

Overall, the paper investigates the research question very systematically and shows experiments that clearly support their findings.

Weaknesses:
- By setting context length to 2, the model is actually forced to find Newton's laws because there isn't really an alternative. The model must compute something like velocity using the difference between the two timesteps and estimate force from acceleration. Using small context lengths such as 3-5 makes more sense, which the paper does show.
- The authors argue that the Newtonian model would be much better for generalization, but I don't see any results to confirm this. In fact, the Keplerian world model achieves lower prediction error than the Newtonian model. One way you could test OOD generalization is to train on orbits of some range of eccentricities and then test on orbits of unseen eccentricity.
- I think some dicussion of physics-informed networks such as Hamiltonian (Greydanus et al. 2019) or Lagrangian neural networks (Cranmer et al. 2020) is warranted. These are highly relevant alternative approaches one could use to learn physical world models, albeit with much stronger inductive biases.

---

> ### Author Rebuttal · Authors · 2026-03-30
>
> We would like to thank the reviewer for their kind words and constructive suggestions. Below we respond to the weakness points.
> * The model is forced to find Newton’s laws when context length is 2.
>     * We share the same intuition, which is the very motivation for us to vary context lengths.
> * The authors argue that the Newtonian model would be much better for generalization.
>     * We apologise for the confusion, and we will revise to tone it down. We do not have evidence for out-of-distribution generalization yet. Learning the correct equation for in-distribution data is a good sign of generalization, but does not guarantee that. The key gaps remain: how to extract the correct equation symbolically (beyond linear probing) and apply the symbolic equation to out-of-distribution data? A subtle point is to distinguish between generalization loss (due to distribution shift in orbit parameters) and robustness loss (due to error accumulation along contexts), and our results show that the Newtonian model has worse robustness loss.
> * Some discussion on related works is warranted.
>     * Thanks for the nice suggestion! We will update the manuscript to include more thorough literature review on “AI Physicists” like HNN or LNN.

---

> > ### Author Rebuttal · Reviewer_5gex · 2026-04-03
> >
> > My concerns are partially resolved. For point 2, I agree that there still remains a gap on showing the benefits of using symbolic equations to OOD data. Also, could the authors answer the question in the Key Questions?

---

### Decision · Program_Chairs · 2026-04-30

**Decision:**

Accept (regular)

**Comment:**

This paper presents an interesting and carefully executed analysis of how inductive biases shape learned world models in transformers. While some of the broader claims should be stated more cautiously, the empirical findings are systematic and insightful, and the rebuttal appropriately clarified these limitations. Overall, I lean toward acceptance.